# Carbonation of a Synthetic CAF Compound by CO_2_ Absorption and Its Effect on Cement Matrix

**DOI:** 10.3390/ma16237344

**Published:** 2023-11-25

**Authors:** Woong-Geol Lee, Seung-Min Kang, Myong-Shin Song

**Affiliations:** 1Research Center of Advanced Convergence Processing on Materials, Kangwon National University, Samcheok 25913, Republic of Korea; woongeol.lee@gmail.com; 2Technical Research Center, KC Green Materials Co., Ltd., Samcheok 25961, Republic of Korea

**Keywords:** net-zero, CO_2_ absorption, accelerated carbonation, CaCO_3_, Fick’s second law

## Abstract

In the field of construction materials, the development of fundamental technologies to reduce energy consumption and CO_2_ emissions, such as manufacturing process improvement and the expanded use of alternative materials, is required. Technologies for effectively reducing energy consumption and improving CO_2_ absorption and reduction that can meet domestic greenhouse gas reduction targets are also required. In this study, calcium–aluminate–ferrite (CAF), a ternary system of CaO·Al_2_O_3_·Fe_2_O_3_, was sintered at a low temperature (1100 °C) to examine the possibility of CO_2_ adsorption, and excellent CO_2_ absorption performance was confirmed, as the calcite content was found to be 11.01% after 3 h of the reaction between synthetic CAF (SCAF) and CO_2_. In addition, the physical and carbonation characteristics were investigated with respect to the SCAF substitution rate for cement (10%, 30%, 50%, 70%, and 100%). It was found that SCAF 10% developed a compressive strength similar to that of ordinary Portland cement (OPC 100%), but the compressive strength tended to decrease as the SCAF substitution rate increased. An increase in the SCAF substitution rate led to the rapid penetration of CO_2_, and carbonation was observed in all the specimens after 7 days. As carbonation time increased, the CO_2_ diffusion coefficient tended to decrease. This is because the diffusion of CO_2_ in the cement matrix follows the semi-infinite model of Fick’s second law. SCAF can contribute to reduced energy consumption and CO_2_ emissions because of the low-temperature sintering and can absorb and fix CO_2_ when a certain amount is substituted.

## 1. Introduction

Since the Paris Climate Change Accord in 2015, countries worldwide have set goals to reduce carbon emissions by >45% compared with 2010 and achieve net-zero by 2050 [1,2]. South Korea declared “net-zero by 2050” and raised the nationally determined contribution (NDC) in 2030 to 40% compared with 2018. According to a press release from the Ministry of Environment (2022), the national greenhouse gas (GHG) emissions in 2021 are expected to be 679.6 million tons, which is an increase of approximately 23 million tons (3.5%) relative to the previous year. The representative industries that generate GHG emissions are the steel, petrochemical, oil refining, and cement industries. They exceed 79% of all industrial GHG emissions and are important for achieving net-zero targets. In particular, the cement industry emitted 0.87 tons of CO_2_ per ton of cement as of 2000, and 0.53 tons of CO_2_ are generated by the decarbonization of limestone [3,4].

In the field of construction materials, it is necessary to develop fundamental technologies to reduce energy consumption and CO_2_ emissions, such as manufacturing process improvement and expanded use of alternative materials, in addition to technologies for effectively reducing energy consumption and improving CO_2_ absorption and reduction that can meet domestic GHG reduction targets. Research has been actively conducted on the use of blast furnace slag, which is a byproduct of the steel industry, and fly ash, which is a byproduct of coal power plants, as cement substitutes [5,6]; however, they fail to develop compressive strength at early ages compared with cement. Concrete carbonation generally reduces the pH of concrete, causing corrosion of the steel reinforcement embedded in the concrete. Therefore, studies have been conducted on the prediction of the concrete carbonation depth and the inhibition of carbonation [7,8,9,10].

Concrete carbonation has effects such as pore-filling in concrete, densification, and compressive-strength improvement because the carbonation product, that is, CaCO_3_, has a higher molecular weight than hydrates [11,12,13]. Therefore, positive effects of carbonation on concrete can be expected in ordinary concrete and precast concrete products. Studies have been conducted on the effects of CO_2_ curing that artificially uses CO_2_. The representative effects of CO_2_ curing include porosity reduction, performance-inhibitor blocking, compressive-strength improvement, shrinkage compensation by carbonation, and CO_2_ reduction due to the reaction with CO_2_. Previous studies were limited to ordinary Portland cement (OPC) and focused on the rate of increase in the compressive strength, as they were conducted in the early stage of research on CO_2_ curing [14,15,16]. C_4_AF, a cement clinker product, underwent a phase change at a sintering temperature of 700–1200 °C and was converted into ferrite in a stable state at the time of cooling after being in an unstable solid solution state with various compositions, such as 2CaO·Fe_2_O_3_, 4CaO·Al_2_O_3_·Fe_2_O_3_, and 6CaO·2Al_2_O_3_·Fe_2_O_3_ [17]. The CaO·Al_2_O_3_·Fe_2_O_3_ ternary system is cubic, and its hydration reaction is divided into the C_3_AH_6_-C_3_FH_6_ series (garnet–hydrogarnet series) and the C_4_AH_n_-C_4_FH_n_ series (hexagonal system); it is known that the hydration reaction by water and the carbonation reaction by CO or CO_2_ is possible.

Therefore, in this study, we investigated the CO_2_ absorption and fixation characteristics of synthetic calcium aluminoferrite (SCAF), which was synthesized at a temperature (1100 °C) lower than the cement sintering temperature (1450 °C), resulting from the carbonation reaction. We further examined the characteristics of the cement matrix and CO_2_ diffusion characteristics with respect to the SCAF substitution rate for cement.

## 2. Materials and Methods

### 2.1. Materials

Table 1 presents the physical and chemical properties of the materials used in this study. For the cement, OPC was obtained from Sampyo cement (Samcheck-si, Republic of Korea) and had a density of 3.15 g/cm^3^ and fineness of 3.475 cm^2^/g. For the sand used in the mortar, ISO standard sand with a SiO_2_ content of ≥90 wt.%, a moisture content of ≤0.2 wt.%, and a fineness modulus of 2.65 was employed.

Figure 1 shows the CAF (mixed in molar ratio; CaO, Al_2_O_3_, Fe_2_O_3_) sintering process and conditions. The mixture was heated at a rate of 10 °C/min until the temperature reached 1100 °C. After 2 h of sintering at a low temperature (1100 °C), the mixture was cooled in a furnace to room temperature and pulverized with a ball mill to produce SCAF.

### 2.2. Methods

#### 2.2.1. Carbonation Characteristics of SCAF

A carbonation test was conducted to evaluate the CO_2_ reactivity of the SCAF. In the test, SCAF was dispersed in water (binder: water = 1:1) to convert it into a slurry state and subjected to a gas–liquid (wet method) reaction with CO_2_, as shown in Figure 2. The reaction time for CO_2_ was set at 30 min, 1 h, 3 h, and 6 h. Each time, samples for analysis were collected, and their crystal structures were examined via a qualitative X-ray diffractometer (Rigaku, D/Max-2500 V, Tokyo, Japan scan range: 5 °–80 ° accelerating voltage: 40 kV, 200 mA; scan speed: 2 °/min; target: Cu) analysis (5 °/min). In addition, the hydrates generated through carbonation were subjected to a thermal analysis using an STA409PC Luxx instrument (Netzsch, Selb, Germany). For the thermogravimetry differential thermal analysis (TG-DTA), the temperature was increased at 10 °C/min from room temperature to a maximum value of 1000 °C, and the calcite content was measured in a nitrogen (N_2_) atmosphere to calculate the amount of adsorbed CO_2_.

#### 2.2.2. Effects of SCAF Carbonation on the Properties of Cement Matrix

A test was conducted to examine the material properties of the cement matrix and CO_2_ diffusion for the matrix with the substitution of SCAF by setting the SCAF substitution rate at 10%, 30%, 50%, 70%, and 100%, as shown in Table 2. The cement-to-sand ratio was set at 1:3, and the water-to-cement (W/C) ratio was set at 0.5.

To investigate the compressive-strength characteristics with respect to the SCAF substitution rate, prismatic specimens with dimensions 40 × 40 × 160 mm were prepared in accordance with ASTM C 109 [18]. The specimens were subjected to dry curing (20 ± 1 °C, 60% ± 5% RH) until their compressive strengths were measured. The flexural strength was first measured according to age, and then the compressive-strength test was conducted using the broken specimens. In addition, the porosity was measured by mercury intrusion porosimetry to examine the pore structures in the hydrates. The carbonation test was conducted in accordance with KS F 2584 [19] (“Standard Test Method for Accelerated Carbonation of Concrete”), and 40 × 40 × 160 mm specimens for carbonation were prepared by setting the water/binder ratio at 0.5, with a sand/binder ratio of 3. The specimens were subjected to water curing at 20 ± 2 °C for 28 d. They were then cured in a constant-temperature and humidity chamber with a humidity of 60% ± 5% and temperature of 20 ± 2 °C. The pouring surface, the bottom surface, and both sides were coated three times using epoxy resin to block CO_2_. The specimens were subjected to carbonation tests using two methods. First, they were cured for different ages (3, 28, and 91 days) in a vacuum desiccator with a humidity of 60% ± 5%, temperature of 20 ± 2 °C, and CO_2_ gas concentration of 100%. Second, they were exposed to the atmosphere, and measurements were performed according to their age (3, 28, and 91 days) to examine the carbo reaction caused by the adsorption of CO_2_ in the atmosphere. Regarding the test surface preparation and measurement method, the specimens were cut to a size of 40 × 40 × 10 mm. The cut surface was sprayed with a 1% phenolphthalein solution and the change in color of the surface to purple was measured in accordance with KS F 2596 [20] (“Method for Measuring Carbonation Depth of Concrete”). Figure 3 shows the accelerated-carbonation experimental setup.

## 3. Results and Discussion

### 3.1. Characteristics of SCAF

The SCAF had a density of 3.52 g/cm^3^, and its fineness after pulverization was 3117 cm^2^/g. Figure 4 shows the X-ray diffraction (XRD)–Rietveld analysis results for the SCAF, which consisted of 50.8 wt.% srebrodolskite (Ca_2_Fe_2_O_5_), 16.5 wt.% mayenite (Ca_12_Al_14_O_33_), 11.5 wt.% krotite (CaAl_2_O_4_), unreacted alumina, and calcium oxide.

### 3.2. Carbonation Characteristics of SCAF

Figure 5 shows the crystalline phase analysis results for the reaction hydrates at different reaction times (30 min, 1 h, 3 h, and 6 h) after the SCAF powder was dispersed in water to convert it into a slurry state and cause the gas–liquid (wet method) reaction with CO_2_. After 3 h of carbonation, numerous hydrates were generated by the reaction between SCAF and CO_2_, including calcite, calcium carboaluminate compounds (CAC(Ca_4_Al_2_O_6_CO_3_·11H_2_O) and CACH(Ca_4_Al_2_O_6_CO_3_·11H_2_O)), and calcium carboaluminoferrite compounds (CFC(Ca_4_A_l2_Fe_2_O_12_CO_3_(OH)_2_·22H_2_O)). In addition, the peak intensity of lime stopped decreasing as the carbonation reaction time increased. This appeared to be because the CaO that remained in the SCAF compounds contributed to the generation of CaCO_3_ by the carbonation reaction. In particular, the peak intensity of CaCO_3_ was strong after 3 h of carbonation. After 6 h of the carbonation reaction, the peak intensity was further increased. The intensities of the peaks corresponding to CAC and CFC crystals, i.e., calcium carbo compounds, increased with the reaction time. This indicated that SCAF generates calcite and calcium carbo compounds through the reaction with CO_2_ and that the carbo reaction becomes more active over time. Because SCAF generates calcite and calcium carbo compounds by adsorbing CO_2_ as a result of the carbo reaction, it is applicable as a CO_2_ absorption and CO_2_ fixation material.

After SCAF was dispersed in water to convert it into the slurry state and cause the gas–liquid (wet method) reaction with CO_2_, sampling was performed according to the CO_2_ reaction time, and TG-DTA was conducted to examine the adsorbed amount of CO_2_. Table 3 presents the weight loss for each of the three temperature sections. The first section corresponds to the weight lost by the decomposition of water, and the second section corresponds to the weight lost by Ca(OH)_2_. Finally, the third section corresponds to the weight loss in the 700–900 °C range, which is the typical thermal decomposition characteristic of calcite generated through the absorption of CO_2_ by the hardened body of SCAF. In the first section, corresponding to the weight lost by the decomposition of H_2_O, the loss of water from the generated compound decreased as the carbonation reaction time increased. The amount of Ca(OH)_2_ that decomposed in the second section also decreased. However, the amount of thermal decomposition in the third section increased. This indicates that the carbonation of SCAF interferes with the absorption of H_2_O or its reaction in the presence of CO_2_. The weight loss in the third section was caused by the thermal decomposition of CaCO_3_ generated by the carbonation reaction, indicating that the amount of CaCO_3_ generated increased with the carbonation reaction time.

According to Duguid et al., supercritical CO_2_ (scCO_2_) and water under certain conditions form a two-phase system composed of wet scCO_2_, a phase rich in scCO_2_, and a water-rich phase (CO_2_-saturated aqueous solution) because they can dissolve in each other without being mixed [21]. Therefore, cementitious materials can be exposed to CO_2_-saturated aqueous solutions, wet scCO_2_, or their combination. In this case, CO_2_ is dissolved in water to form carbonic acid, which forms calcium carbonate in cementitious materials. The CAF used in this study was a cementitious material, and it was determined that calcium carbonate was generated by the carbonation reaction of SCAF owing to exposure to the CO_2_-dissolved aqueous solution in the carbonation reaction, even though it was not supercritical CO_2_.

### 3.3. Effects of SCAF Carbonation on the Cement Matrix

#### 3.3.1. General Properties

The compressive strengths at each age (3, 7, and 28 days) were measured and compared for the reference OPC mixture (P) and SCAF substitution rates of 10%, 20%, 30%, 50%, 70%, and 100%, as shown in Figure 6.

As the SCAF substitution rate increased, the compressive strength at each age tended to decrease. SCAF 10% developed a similar strength to 100% OPC (P). However, at substitution rates of 20%, 50%, and 70%, the compressive strength ranged from 50% to 2.3% of that of 100% OPC at 3 days of age and from 70% to 2.3% at 28 days. In the case of SCAF 100%, almost no strength developed at 3 and 28 days of age. Thus, the compressive strength decreased rapidly as the SCAF content increased. This indicates that the strength development due to the hydration reaction is insignificant for SCAF and that SCAF has no hardening characteristics due to the hydration reaction. To evaluate the pore structure characteristics with respect to the SCAF content, the pore-size distribution at 28 days of age was analyzed for the 100% OPC, SCAF 10%, and SCAF 100% specimens. The porosity of the cement was 23.76% at 28 days, whereas those of the SCAF 10% and SCAF 100% specimens were higher (40.12% and 52.84%, respectively).

Figure 7 shows the pore-size distributions for each specimen. SCAF 100% contained more capillary pores (≥50 nm) and fewer mesopores (4.5–50 nm) and gel micropores (3.0–4.5 nm) than the 100% OPC and SCAF 10% specimens. SCAF 100% exhibited a high porosity owing to the evaporation of a large amount of residual water that could not contribute to the hydration reaction. The presence of such pores facilitates the penetration or diffusion of CO_2_ during the carbonation reaction.

#### 3.3.2. Penetration and Diffusion of CO_2_ in Cement Matrix Containing SCAF Compounds

After the carbonation test, the area of the specimen was largely divided into two areas: (1) a colorless area in which carbonation occurred along with the consumption of hydroxyl ions when Ca(OH)_2_ reacted with CO_2_ to form CaCO_3_, and (2) a purple area without a carbonation reaction in which hydroxyl ions that did not react with existing CO_2_. The carbonation depth by age was measured for each specimen, and the results are presented in Table 4.

Carbonation was performed using an accelerated method. Carbonation hardly occurred in the 100% OPC (P) specimen and in the cases where a small amount of SCAF was added. As the SCAF content increased, rapid carbonation occurred at an early stage. For the ordinary cement with no SCAF and the cement matrix containing a small amount of SCAF compounds, it appeared that the penetration or diffusion of CO_2_ due to accelerated carbonation was hindered because the density of the matrix was increased by the generation of calcium silicate hydrates, calcium aluminate hydrate, ettringite, or monosulfate owing to the active hydration of cement. This formed a two-phase system composed of wet scCO_2_, a phase rich in scCO_2_, and a water-rich phase (CO_2_-saturated aqueous solution), as reported by Duguid et al. [21]. Therefore, according to the theory that cementitious materials can be exposed to a CO_2_-saturated aqueous solution, wet scCO_2_, or both, in which CO_2_ is dissolved in water to form carbonic acid, and that carbonic acid forms calcium carbonate in cementitious materials, it is judged that hardened cement resists the penetration and diffusion of CO_2_ in the early stage of carbonation, but occur through the pores in hardened cement after a certain period of time. When the SCAF substitution rate was ≥30%, CO_2_ penetrated more rapidly as the substitution rate increased, and all the specimens were carbonated at 7 days of age. In the case of accelerated carbonation, carbonation occurred rapidly because the CO_2_ concentration was high. In general, cement hydrates are carbonated through reactions with CO_2_ in the atmosphere, and the types of reactions are presented in Table 5. For these reactions, carbonation occurs after the hydration of cement, indicating that the penetration and diffusion of CO_2_ in the cement matrix proceed after the occurrence of hydration at a certain age.

#### 3.3.3. CO_2_ Diffusion Coefficient Calculation

The theory of diffusion follows Fick’s law. Fick’s second law describes the changes in the concentration of the diffusing solute at a position over time based on the first law. Therefore, it considers changes in the concentration of a sample at a given position over time. As for diffusion in the cement matrix where carbonation proceeds, the concentration increase in the unit area over time can be explained by the difference between the inflow and outflow volume fluxes according to Equation (1). Here, *F_in_* represents the mass of incoming CO_2_, and *F_out_* represents the mass of outgoing CO_2_. Δ*C* represents the amount of calcium carbonate (excluding CaO), combined during carbonation.
(1)$$F_{in}=F_{out}+∆C$$

Fick’s second law quantifies diffusion in a specimen as the change in concentration over time. The concentration of the diffusion component decreases as the distance from the surface (*x*) increases, and increases over time at a given position of *X*. The mathematical solution to Fick’s second law can be expressed using Equation (2), which is limited by the boundary condition. The concentration at time *t*(*C*(*x*, *t*)) at a given distance *x* can be calculated using Equation (3), which uses the error function for diffusion in a semi-infinite solid.
(2)$$\frac{dC}{dt}=D\frac{d^{2}C}{{dX}^{2}}$$
(3)$$\frac{C_{x}}{C_{0}}=1-erf\left(\frac{X}{\sqrt[{2}]{Dt}}\right)$$

When the surface concentration *C*_0_ and the concentration at distance *X* (*C_X_*) are known, the diffusion coefficient *D* can be calculated using Equation (4).
(4)$$X^{1}=k\sqrt{Dt}$$

Table 6 presents the diffusion coefficients according to the CO_2_ penetration depth. As shown in the figure, the diffusion coefficient was large in the early stage of carbonation but decreased over time. This indicates that the diffusion of CO_2_ in the cement matrix closely follows a semi-infinite model of Fick’s second law. Thus, the diffusion of CO_2_ due to penetration is easy in the early stage of carbonation, but it is slowed by the filling of pores by carbonation products over time.

According to these research results, SCAF can contribute to a reduction in CO_2_ emissions through energy savings because it can be sintered at a lower temperature (1100 °C) than cementitious compounds or cement (1450 °C). SCAF is also expected to contribute to the absorption and fixation of CO_2_ when a certain amount is substituted. The use of CAF compounds may cause degradation of properties, such as the compressive strength, but this can be improved through optimal mix design.

## 4. Conclusions

We investigated the carbonation potential of SCAF together with the physical and carbonation characteristics according to the substitution of SCAF for cement, and the following results were obtained.The SCAF was prepared via low-temperature sintering (1100 °C). Its density was 3.52 g/cm^3^ and its fineness after pulverization was 3117 cm^2^/g. The main crystalline phases were srebrodolskite (Ca_2_Fe_2_O_5_), mayenite (Ca_12_Al_14_O_33_), krotite (CaAl_2_O_4_), unreacted alumina, and CaO.When wet carbonation was performed to evaluate the carbonation potential of SCAF, the calcite content rapidly increased to 11.01% after 3 h of the carbonation reaction, confirming that SCAF has a CO_2_ absorption and fixation effect.When SCAF was substituted for 10% of cement, the compressive strength was equal to that of the 100% OPC (P) specimen. However, as the SCAF substitution rate increased, the compressive strength tended to decrease.SCAF 100% contained more capillary pores and fewer mesopores and gel micropores than the 100% OPC (P) and SCAF 10% specimens. CAF 100% exhibited a high porosity owing to the evaporation of a large amount of residual water that could not contribute to the hydration reaction.Carbonation hardly occurred in the 100% OPC (P) specimen and in cases where a small amount of SCAF was added. However, as the SCAF content increased, rapid carbonation occurred at an early stage.The diffusion coefficient decreased as the carbonation time increased. This indicated that the diffusion of CO_2_ in the cement matrix closely follows a semi-infinite model of Fick’s second law.

SCAF can contribute to reducing energy consumption and CO_2_ emissions because of the low-temperature sintering and can absorb and fix CO_2_ when a certain amount is substituted. Further research is needed to evaluate the durability of the cement matrix with the substitution of SCAF for cement.

## Figures and Tables

**Figure 1 materials-16-07344-f001:**
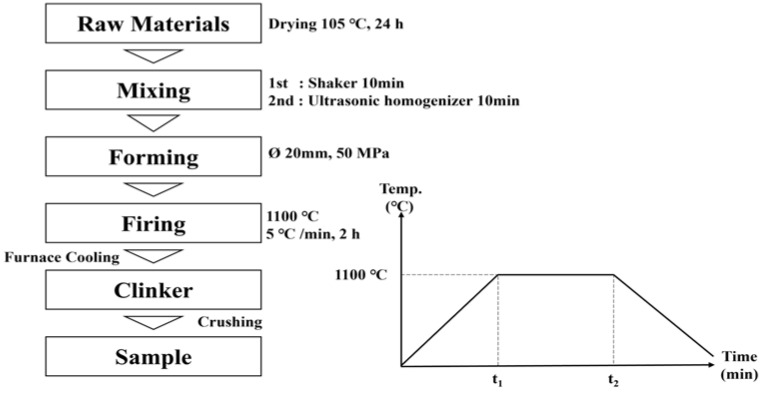
SCAF sintering process.

**Figure 2 materials-16-07344-f002:**
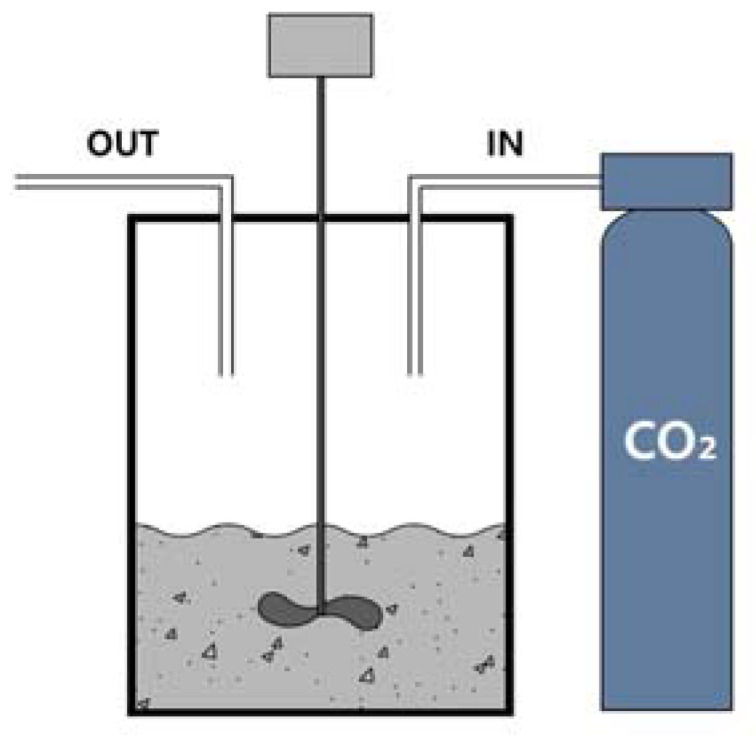
Carbonation method.

**Figure 3 materials-16-07344-f003:**
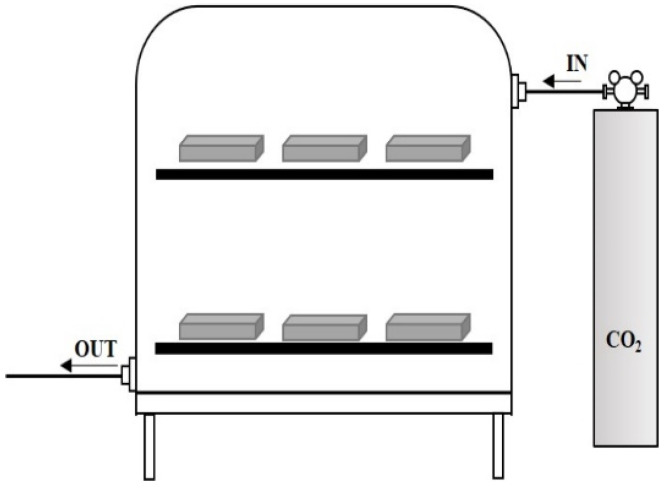
Accelerated-carbonation experimental system.

**Figure 4 materials-16-07344-f004:**
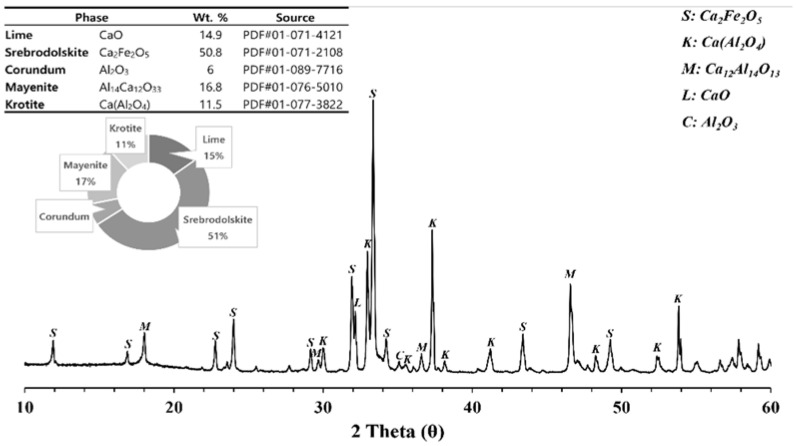
XRD pattern of the SCAF.

**Figure 5 materials-16-07344-f005:**
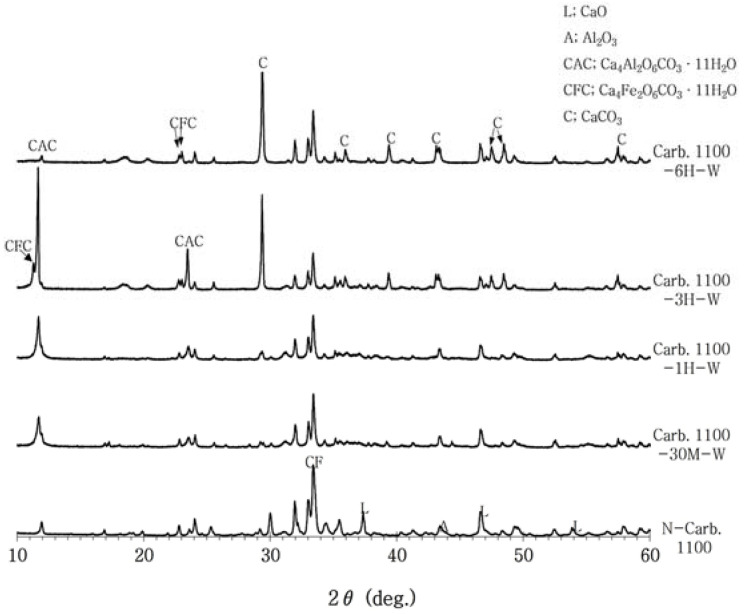
XRD patterns for carbonation with the wet method.

**Figure 6 materials-16-07344-f006:**
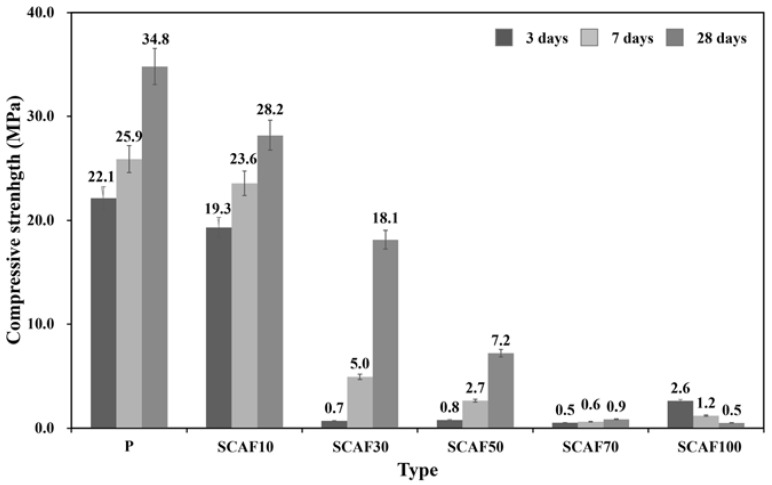
Compressive strength with respect to the SCAF substitution rate.

**Figure 7 materials-16-07344-f007:**
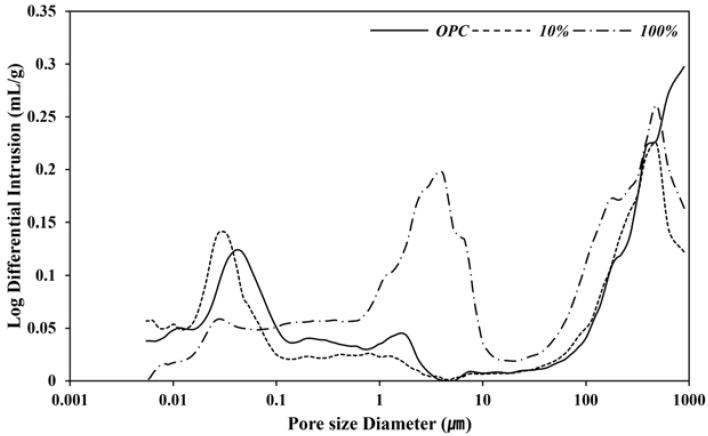
Pore-size distribution of paste with SCAF.

**Table 1 materials-16-07344-t001:** Chemical and physical properties of materials.

	Component (wt.%)	Materials	
		OPC	SCAF
Chemical	SiO_2_	20.5	3.38
	Al_2_O_3_	5.0	10.49
	Fe_2_O_3_	3.4	30.44
	CaO	62.3	53.0
	MgO	3.6	0.5
	SO_3_	2.1	1.49
	Loss on ignition	2.4	·
Physical	Blaine (cm^2^/g)	3475	3117
	Specific gravity (g/cm^3^)	3.15	3.52

**Table 2 materials-16-07344-t002:** Mixing design.

Sample No.	Label	Binder (%)		Paste Ratio	
		OPC	SCAF	Sand/Binder	Water/Binder
1	P	100	0	3	0.5
2	SCAF10	90	10		
3	SCAF30	70	30		
4	SCAF50	50	50		
5	SCAF70	30	70		
6	SCAF100	0	100		

**Table 3 materials-16-07344-t003:** Weight loss for different temperature sections according to the carbonation time.

Reaction Time	First Section 100–200 °C	Second Section 350–500 °C	Third Section 600–900 °C
30 min	6.32	7.27	2.82
1 h	5.89	7.14	4.28
3 h	4.97	6.34	11.01
6 h	2.27	6.12	12.93

**Table 4 materials-16-07344-t004:** Carbonation depth (mm) in accelerated carbonation.

	Accelerated Carbonation		
	3 Days	28 Days	91 Days
P	0	0	0
SCAF10	5.21	5.41	0
SCAF30	10.15	40.00	40.00
SCAF50	35.21	40.00	40.00
SCAF70	40.00	40.00	40.00
SCAF100	40.00	40.00	40.00

**Table 5 materials-16-07344-t005:** Carbonation in cement hydration [22].

1. Carbonation of calcium silicate hydrate 3CaO·SiO_2_·3H_2_O+CO_2_→3CaCO_3_+2SiO_2_+3H_2_O
2. Carbonation of calcium hydroxide Ca(OH) _2_→CaCO_3_+H_2_O
3. Carbonation of ettringite 3CaO·Al_2_O_3_·3CaSO_4_·32H_2_O+3CO_2_→3CaCO_3_+2Al(OH)_3_+3CaSO_4_·H_2_O+32H_2_O
4. Carbonation of fredelitis 3CaO·Al_2_O_3_·CaCl_2_·10H_2_O+3CO_2_→3CaCO_3_+2Al(OH)_3_+3CaCl_2_+29H_2_O

**Table 6 materials-16-07344-t006:** Carbonation coefficient in accelerated carbonation.

	Curing Age		
	3 Days	28 Days	91 Days
P	0.00	0.00	0.00
SCAF10	7.96	2.71	0.00
SCAF30	15.50	20.00	11.09
SCAF50	53.78	20.00	11.09
SCAF70	61.10	20.00	11.09
SCAF100	61.10	20.00	11.09

## Data Availability

Data are contained within the article.

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
