# Peer review of "Carbonation of a Synthetic CAF Compound by CO2 Absorption and Its Effect on Cement Matrix"

_materials, 2023, doi:10.3390/ma16237344_

Round 1

Reviewer 1 Report

Comments and Suggestions for Authors

This paper deals with interesting topic in the field of construction materials, to development technologies to reduce energy consumption and CO2 emissions. The authors conducted valuable research where calcium-aluminate-ferrite a ternary system of CaO·Al2O3·Fe2O3 was sintered at a low temperature to examine the possibility of CO2 absorption. The experimental results obtained demonstrated that low values of SCAF can be used to generate materials with CO2 embodied with a same compressive strength that plain cements. The paper was well structured, and a minor revision is required.

Author Response

Thank you for taking the time to review my paper. I have thought more deeply about my thesis thanks to your advice, and I think the quality of my thesis has improved. Thank you again.

Reviewer 2 Report

Comments and Suggestions for Authors

In the manuscript, the characterization of calcium-aluminate-ferrite (CAF) samples, a ternary system of CaO·Al2O3·Fe2O3, has been carried out at low temperature (1,100 °C), using a common characterization technique. The analyzed samples are intended to examine the possibility of CO2 adsorption.

In addition, the physical and carbonation characteristics were investigated with respect to the SCAF substitution rate for cement (10%, 30%, 50%, 70% and 100%).

Although the paper fits the scope and objectives of the journal proposed for publication, the paper must undergo major review before further evaluation.

The main drawbacks are the following:

In the title they put Synthetic CAF when in the rest of the work they use the SCAF abbreviation to refer to the synthesized CAF.

Sometimes they use spaces between the numerical value and the units, and sometimes they don't. Please normalize this and always respect the same system.

Line 21. 7 d should be written as 7 days.

Line 53. There is a writing error in references 11 and 13.

Line 58. When a name is abbreviated, the beginning of each word must be capitalized. Therefore, it should be written as Ordinary Portland Cement.

Line 58. The part from "Previous studies..." begins to complicate the reading and it is not clear why it names all those compounds. I understand that it has something to do with the study and transformation of materials, but I think it should be explained better and give more information. That there is a coherence in the reading.

Line 69. The authors should better indicate how the synthesis of SCAF has been carried out and comment on the synthesis method used.

Table 1. What does L.O.I mean?

Line 78. Remove the comma before the "and".

Line 91. What do the authors mean by wet method? If they have used that method, they should explain what the method implies, the advantages and why this method has been selected and not another.

Table 2. In the label part, indicate that what is written is in percentage.

CAF or SCAF?

Figure 3. The authors should indicate in more detail what happens in the process. It would look good if they put some information inside the image.

Line 137. The density and fineness that is named. is it referring to which sample? The SCAFs have been synthesized with different percentages and it is not clear to which sample they refer. Explain it.

Figure 4. Both the table and the figure included in the X-rays look bad. On the other hand, I suggest that all the X-Rays of all the SCAF synthesized with different percentages be included so that the differences between all the samples used in this work are clearly seen.

Line 147. What happens before 3 hours of carbonation? Do not miss anything? Explain it.

Figure 5. Names need to be changed. Units must be in lowercase.

Lines from 163 to 178. It would be good to show it in the form of a graph (a figure) indicating each stage and what is happening in each of them. It would be much more illustrative and it would be more comfortable and easy for the reader.

Line 184. It is not clear what they mean regarding scCO2. Are the authors saying that what is produced has characteristics and properties similar to those of scCO2?

Lines from 211 to 216. They really do not explain much of the porosity results obtained. They should reasonably comment on the results and explain why they are due. It must be clear. Please, indicate possible reasons behid the phenomenon discussed  in this paragraph.

Table 5. Are the reactions described irreversible? If not, authors should be careful in their use of arrows.

Figure 9. Compound names don't look right.

Finally, I think that SEM images of the samples should be included in order to see the structure of the samples before and after CO2 diffusion. This would give valuable information and would improve the understanding of what happens in the diffusion of CO2.

How much is the deterioration of the carbonation activity? Please, specify the approximate value.

Author Response

(The authors gave the same response as above.)

Reviewer 3 Report

Comments and Suggestions for Authors

This manuscript addresses the carbonation of synthetic CAF by CO2 absorption and the effect of its inclusion on cement. It is and interesting and relevant topic, but the manuscript is not always clear. Some sentences appear without context and others are difficult to understand.

1.     Line 14: maintain consistency when using thousand separators. Always use or never use.

2.     According to SI, a space should be included before the unit symbol, including %

3.     Line 19: Plain was not yet defined. Perhaps “Plain cement”.

4.     Line 21: “The CO2 diffusion coefficient decreased as the carbonation time.” Incomplete sentence.

5.     Line 36: “They >79% of”. Incomplete sentence.

6.     Line 53: “[1113]” the authors refer to references 11-13. An “-“ is missing.

7.     Line 60: “…conducted in the early stage of research on CO2 curing.” please include references for these works.

8.     Lines 60-67: the text is not clear. Please rephrase.

9.     Table 1, SCAF column: is this the composition for the SCAF raw materials? It is not clear which mixture is used as raw material.

10.  Lines 97 and 165: DT-TGA was not defined, only TG-DTA.

11.  Line 107 and table 2: “Binder:Sand”, “Water/Binder”. Both are ratios, maintain consistency.

12.  Line 114: “…was measured mercury…”. Check the sentence. “by” missing?

13.  Ine 123: Two methods or two steps? 

14.  Line 129: not clear. Cut, apply phenolphthalein, measure depth.

15.  Line 146: “caus” ???

16.  Figure 6: legend missing

17.  Figure 7: please check units and description in lines 211-213 

18.  Line 218: not clear

19.  Figure 8: CAF10 series does not represent the values from table 4. Y-axis label: replace deep by depth.

20.  Line 218: context is missing

21.  Line 236: Duguid et al. reference missing

22.  Line 243: “arbonation”?

23.   Equation 2 should be a partial differential equation

24.  Line 323: “Comparison of Comparison of environmental” there might typo here

Author Response

(The authors gave the same response as above.)

Round 2

Reviewer 2 Report

Comments and Suggestions for Authors

The paper is ready to be published